# Optimal Expression, Function, and Immunogenicity of an HIV-1 Vaccine Derived from the Approved Ebola Vaccine, rVSV-ZEBOV

**DOI:** 10.3390/vaccines11050977

**Published:** 2023-05-12

**Authors:** Hiva Azizi, Jason P. Knapp, Yue Li, Alice Berger, Marc-Alexandre Lafrance, Jannie Pedersen, Marc-Antoine de la Vega, Trina Racine, Chil-Yong Kang, Jamie F. S. Mann, Jimmy D. Dikeakos, Gary Kobinger, Eric J. Arts

**Affiliations:** 1Département de Microbiologie-Infectiologie et Immunologie, Faculté de Médecine, Université Laval, Québec, QC G1V 0A6, Canada; hi.azizi@gmail.com (H.A.); alice.berger.1@ulaval.ca (A.B.); marc-alexandre.lafrance.1@ulaval.ca (M.-A.L.); jannie.pedersen.1@ulaval.ca (J.P.); marc-antoine.delavega@crchudequebec.ulaval.ca (M.-A.d.l.V.); trinadracine@gmail.com (T.R.); 2Human Health Therapeutics, National Research Council Canada, Ottawa, ON K1N 5A2, Canada; 3Department of Microbiology and Immunology, Western University, London, ON N6A 3K7, Canada; jknapp03@uoguelph.ca (J.P.K.); yli3685@uwo.ca (Y.L.); cykang@uwo.ca (C.-Y.K.); jimmy.dikeakos@uwo.ca (J.D.D.); 4Galveston National Laboratory, Department of Microbiology and Immunology, University of Texas Medical Branch, Galveston, TX 77555, USA; gary.kobinger@crchudequebec.ulaval.ca; 5Vaccine and Infectious Disease Organization, University of Saskatchewan, Saskatoon, SK S7N 5E3, Canada; 6Bristol Veterinary School, University of Bristol, Langford House, Langford, BS40 5DU Bristol, UK; j.mann@imperial.ac.uk

**Keywords:** human immunodeficiency virus type 1 (HIV-1), vesicular stomatitis virus (VSV) vector, HIV-1 Envelope glycoprotein, Ebola virus glycoprotein

## Abstract

Vesicular stomatitis virus (VSV) remains an attractive platform for a potential HIV-1 vaccine but hurdles remain, such as selection of a highly immunogenic HIV-1 Envelope (Env) with a maximal surface expression on recombinant rVSV particles. An HIV-1 Env chimera with the transmembrane domain (TM) and cytoplasmic tail (CT) of SIVMac239 results in high expression on the approved Ebola vaccine, rVSV-ZEBOV, also harboring the Ebola Virus (EBOV) glycoprotein (GP). Codon-optimized (CO) Env chimeras derived from a subtype A primary isolate (A74) are capable of entering a CD4+/CCR5+ cell line, inhibited by HIV-1 neutralizing antibodies PGT121, VRC01, and the drug, Maraviroc. The immunization of mice with the rVSV-ZEBOV carrying the CO A74 Env chimeras results in anti-Env antibody levels as well as neutralizing antibodies 200-fold higher than with the NL4-3 Env-based construct. The novel, functional, and immunogenic chimeras of CO A74 Env with the SIV_Env-TMCT within the rVSV-ZEBOV vaccine are now being tested in non-human primates.

## 1. Introduction

Despite effective antiretroviral therapy, pre-exposure prophylaxis, and topical microbicides, 1.7 million new HIV infections still occur each year, which could be prevented by a protective vaccine [1,2,3]. After 37 years of HIV-1 vaccine research, a maximal 33% protection rate was observed in one trial: an RV144 trial involving a Thai population vaccinated with a canarypox vector (ALVAC) expressing HIV gag/pol/nef and followed by boosting with a recombinant HIV gp120 [4]. The major correlates of protection appeared to be non-neutralizing antibodies against the V1/V2 region of the envelope protein, while broadly neutralizing Abs (bNAbs) were rarely observed [5]. The modest success of RV144 does not discount the importance of bNAbs in protection, considering that the administration of bNAbs to macaques has provided the best immune-associated protection from SHIV infection to date [6,7,8]. As such, the induction of bNAbs or binding antibodies facilitating ADCC remains at the forefront of the HIV vaccination field.

As the only HIV protein visible on the surface of infected cells and virus particles, the HIV-1 Env glycoprotein has evolved numerous tactics to evade immune recognition, which also act as hurdles for the induction of bNAbs. One of the main evasive tactics is the maintenance of low levels of Env on the surface of infected cells and virus particles (10–20 per particle) [9,10,11]. In comparison, the highly related SIV has much higher levels of Env on the surface of infected cells and virus particles (~100 per particle) and is rarely pathogenic in its native primate host [9,12]. The HIV Env cytoplasmic tail (CT) domain has evolved numerous trafficking signals to allow the tight regulation of the subcellular localization of Env [10]. For example, HIV Env is transiently transported to the plasma membrane of infected cells before undergoing endocytosis, thereby minimizing exposure to the extracellular environment [10,13]. Only upon interaction with HIV Gag to form HIV progeny does Env remain on the cell surface, albeit still at relatively low levels. In addition to plasma membrane localization, the HIV Env signal peptide remains inefficient, resulting in a slow rate of Env synthesis and egress through the ER and Golgi [14,15,16]. These characteristics of the HIV Env protein have plagued multiple HIV vaccine efforts.

Herein, the HIV-1 Env gp140 ectodomain (EC) with various transmembrane and intracytoplasmic domains (TMIC) is cloned into the VSV vector for the optimal Env immunogen expression and to develop future HIV-1 vaccine studies. VSV typically infects livestock, such as cattle, horses, and swine, with infrequent asymptomatic infections occurring in humans [17]. As a result, low frequencies of VSV immunity persist in the human population, thereby establishing VSV vectors as a suitable replication-competent vaccine platform [18,19,20,21,22]. Like many Rhabdoviruses, VSV can accommodate multiple foreign genes into its genome without a significant loss of infectivity and incorporates foreign glycoproteins onto the viral particle surface based on VSV assembly and budding from the plasma membrane [23,24]. Immunization with VSV vectors expressing intracellular foreign antigens or propagating the viral vector harboring foreign surface glycoproteins have induced strong humoral and cellular immunity to expressed foreign antigens [19,20]. Most notably, the VSV platform has been used in the generation of the rVSV-ZEBOV vaccine, where the VSV glycoprotein (VSV-G) was replaced with the Ebola virus glycoprotein (EBOV GP) [18,19,25].

VSV has been used as a vector for a possible HIV vaccine. The low expression of HIV Env on pseudotyped VSV [20,26] was improved with the removal of the HIV-1 Env CT domain and enabled the enhanced incorporation of HIV Env on VSV particles. Building upon these studies, a chimeric HIV-1 Env protein, containing the transmembrane (TM) and CT domains of VSV-G, has even greater levels of incorporation on rVSV particles compared with the CT-truncated Env, while retaining proper trimeric conformation [20]. That VSVΔG-Env.BG505 vaccine candidate results in 67% protection in a primate model of infection [27]. Despite conferring impressive protection, the vaccine vector shows less protection in subsequent studies and has some drawbacks [21]. For example, the VSV∆G vector induces only anti-Env-binding antibodies with minimal neutralizing antibodies, is difficult to propagate in vitro, and is reliant on the infection of CD4+/CCR5+ cells for vector expansion in vivo [21,28,29].

To optimize HIV-1 Env immunogens within a VSVΔG HIV vectors, we compare the expression of HIV-1 Env chimeras made with the transmembrane and cytoplasmic tails of VSV_Indiana_ G, Ebola_Zaire_ GP, and SIVmac239 Env on the surface of transfected cell lines or on the VSV vector [20,24,26]. Although these Env chimeras are functional for host-cell entry through CD4 and a co-receptor (CXCR4 or CCR5), vector propagation is dependent on the Ebola GP in place of the VSV G in this system. Aside from the addition of the HIV-1 Env chimera, the VSV vector template is identical to the rVSV-ZEBOV vaccine, approved for human use [25,30]. As described herein, these rVSV-ZEBOV vectors expressing HIV-1 Env chimeras result in high levels of Env-binding antibodies and neutralizing antibodies in immunized mice.

## 2. Materials and Methods

### 2.1. Ethics Statement

There are no human samples or subjects involved in this study. All experiments involving animals were reviewed and approved (protocol 2018-174) by the Animal Use Subcommittee of the University Council on Animal Care (UCAC) for Western University. All approved protocols followed the guidelines set by the Canadian Council on Animal Care (CCAC). UCAC at Western University is also licensed by the CCAC.

### 2.2. Cell Lines

HEK-293T, HeLa, Vero E6 cells (ATCC, Manassas, VA, USA), and TZM-bl cells (NIH AIDS Reagent Program, cat#8129) were maintained in Dulbecco’s modified Eagle medium (DMEM, Wisent, Quebec, QC, Canada) supplemented with 10% FBS (Wisent) and 100 µg/mL of Penicillin-Streptomycin (ThermoFisher Scientific, Waltham, MA, USA). All cells were grown at 37 °C in the presence of 5% CO_2_ and cultured according to suppliers’ recommendations. The following reagent was obtained through the NIH AIDS Reagent Program, Division of AIDS, NIAID, NIH: TZM-bl cells (Cat#8129) from Dr. John C. Kappes, and Dr. Xiaoyun Wu [31,32,33].

### 2.3. VSV Vectors and Chimeric HIV-1 Env Proteins Construction

The use of three VSV vectors, VSVXN2, VSVXN2ΔG/EBOVGP, and VSVXN2ΔG/CDO-EBOVGP are investigated in this study (Figure 1). The VSVXN2 vector contains the full-length VSV genome with the unique MluI and AvrII restriction enzyme sites flanked by VSV transcriptional start/stop signals located between the VSV matrix (M) and VSV glycoprotein (G) genes [34]. The VSVXN2ΔG/EBOVGP and VSVXN2ΔG/CDO-EBOVGP vectors differ from VSVXN2 in that the VSV glycoprotein (G) gene is replaced with either the wild type Makona B6 Zaire EBOV GP gene or a codon de-optimized (CDO) EBOV GP gene with decreased expression [35]. All vectors contain the following elements: bacteriophage T7 promoter, VSV leader sequence, hepatitis delta virus ribozyme sequence, and T7 terminator sequence to allow for the rescue of rVSV particles with the co-transfection of the VSV helper plasmids (pCAGGS-T7, pBS-N, pBS-P, and pBS-L), as previously described [36,37,38].

To construct the VSVXN2ΔG/CDO-EBOVGP vector, modified EBOV GP genes with decreased levels of expression, were synthesized through codon de-optimization. Specifically, the addition of rare codons and negative cis-regulatory elements were introduced. The wild type EBOV GP gene was then replaced with the codon de-optimized EBOV GP gene via In-Fusion cloning.

To produce HIV-1 Env genes with enhanced expression on the surface of VSV-infected cells and rVSV particles, the transmembrane (TM) and cytoplasmic tail (CT) domains of either HIV-1NL4-3 env (GenBank: AFN70961.1) or HIV-1A74 env [39] were replaced with the TM and CT domains of SIVmac239 env with the 18T C-terminal truncation (SIVmac239-T) (GenBank: AAA47637.1), EBOVZaire GP (GenBank: AAB81004.1), or VSVIndiana G (GenBank: M11048.1) genes. For simplicity, only the HIV-1 strain names A74 and NL4-3 will be identified below. As a comparison for chimeric Env incorporation into rVSV particles, the HIV Env mutant, Envtr3 (referred to as Envt713 in our study), which is efficiently incorporated into rVSV particles, was utilized [26]. The Envt713 mutant generated in our study utilized HIV NL4-3 Env and similarly had the majority of the HIV Env CT deleted, leaving only the first three amino acids (RVR) after the TM region. To further optimize expression of chimeric A74 env genes, chimeric genes were synthesized with a mammalian codon bias [40]. Additionally, the signal sequence of codon-optimized A74 chimeric genes was replaced with the signal sequence of the honeybee melittin protein [16,41]. Chimeric Env genes were generated either through PCR amplification with reverse primers containing the TM and CT of the other viral glycoprotein or synthesized via GenScript (Piscataway, NJ). Chimeric HIV-1 Env genes were then cloned into pcDNA3.1, VSVXN2, and/or VSVXN2ΔG/EBOV_GP vectors via In-Fusion cloning to allow for characterization of chimeric Env protein expression on cell surfaces and rVSV particles. To simplify construct names and to emphasize the different HIV-1 Env chimeras utilized, the VSVXN2 and VSVXN2ΔG/EBOV_GP vector names were simplified to rVSV_G and rVSVΔG+EBOV_GP. We also refer to the latter as analogous to the rVSV-ZEBOV vector approved as the Ebola vaccine for human use [41].

For experiments investigating the effect of HIV-1 Rev co-expression on COA74 Env expression, a chimeric COA74 Env containing the TM and CT domains of VSV-G with the HIV Rev response element (RRE) kept intact, COA74_Env-ECRRE/VSV_GTMCT, was generated.

### 2.4. Rescue of Recombinant VSV Vectors

Rescue of rVSV vectors expressing chimeric HIV Env was performed using methods previously described, except that Vero E6 cells and HEK-293T cells were utilized together [36,37,38]. In brief, 1.5 × 10^5^ HEK-293T and Vero E6 cells were plated together in a 6-well plate (3 × 10^5^ cells in total). At 70% confluency, cells were transfected with VSV helper plasmids and the rVSV genome (2 µg rVSV genome, 2 µg T7-pCAGGS, 1.25 µg pBS-VSV-P, 0.5 µg pBS-VSV-N, and 0.25 µg pBS-VSV-L). Following the detection of cytopathic effects (CPE), virus-containing supernatants were collected and used to infect Vero E6 cells to allow for viral propagation. Following the rVSV propagation, virus-containing supernatants were collected, and cellular debris was removed via centrifugation at 500× *g* and 4 °C for 5 min. Following titering on Vero E6 cells, samples were analyzed for protein expression by Western blot.

### 2.5. Mice Immunogenicity Study

Seven 6–8 week-old female BALB/c mice per test group received an intraperitoneal (IP) inoculation with 20 µL PBS containing 10^6^ PFU of rVSV particles. The five groups received immunizations with the following rVSV: rVSVΔG+EBOV_GP, rVSV_G+NL4-3t713_Env, rVSVΔG+EBOV_GP+NL4-3t713_Env, rVSV_G+NL4-3_Env-EC/SIV_Env_TMCT, and rVSVΔG+EBOV_GP+NL4-3_Env-EC/SIV_Env-TMCT, respectively. At day twenty-eight, blood was collected via cardiac puncture and the serum was isolated. All experiments were monitored and performed in compliance with the Laval University Council on Animal Care. A second set of mouse immunizations was performed with the constructs as described below.

### 2.6. Analysis of Env Nucleotide Sequences Using GenScript Rare Codon Analysis Tool

To investigate the suitability of Env coding sequences for expression in a human context, the nucleotide sequences of HIV NL4-3 (GenBank: AFN70961.1) and A74 Env (GenBank: JX993970.1) were analyzed using the GenScript rare codon analysis tool. Analyses provided the overall codon adaptation index (CAI) value [42], which represents how well a specific gene will be expressed in a particular host organism. CAI values can range from 0.0–1.0, with a value in the range of 0.8–1.0 considered ideal. In addition to the CAI value, the number of negative cis regulatory elements (CRE), which refer to sequence motifs that negatively regulate the gene expression at the transcriptional and translational levels, were reported. The GenScript rare codon analysis tool was accessed at https://www.genscript.com/tools/rare-codon-analysis (accessed on 5 May 2019).

### 2.7. Western Blot Analysis

For the analysis of the total cellular expression of chimeric HIV-1 Env proteins, 10 cm dishes containing HEK-293T cells at 70% confluency were transfected with 4 μg of chimeric HIV Env expression vectors and either 1 μg of HIV Rev expression vector (pRSV-Rev) or empty pcDNA3.1 vector. The pRSV-Rev was a gift from Didier Trono (Addgene plasmid # 12253; http://n2t.net/addgene:12253 (accessed on16 January 2023); RRID:Addgene_12253). After 24 h, cells were washed with 1X PBS, followed by lysis (0.5 M HEPES, 1.25 M NaCl, 1 M MgCl2, 0.25 M EDTA, 0.1% Triton X-100, 1X complete protease inhibitor; Roche, Indianapolis, IN). Cells were placed on a rotator for 20 min at 4 °C before scraping and removing insoluble cellular debris by centrifugation at 20,000× *g* for 20 min. Cell lysates were boiled at 93 °C in 5X SDS-PAGE sample buffer (0.312 M Tris pH 6.8, 25% 2-Mercaptoethanol, 50% glycerol, 10% SDS) for 15 min and stored at −20 °C. Proteins were separated via 8% SDS-PAGE gel and subsequently transferred to nitrocellulose membranes. Membranes were blocked in 5% non-fat skimmed milk (BioShop Canada, Burlington, ON, Canada) in 1X TBST, containing 0.1% Triton X-100, for 1 h at room temperature. This was followed by an overnight incubation at 4 °C with primary antibodies: clarified hybridoma supernatant containing mouse anti-gp120 (B13) monoclonal IgG (provided by George Lewis, Institute of Human Virology, Baltimore, MD, and Bruce Chesebro, NIAID, Hamilton, MT, USA) and anti-β actin mAb (1:3000; ThermoFisher Scientific). Membranes were then washed with 1X TBST and incubated for 1.5 h with the appropriate species-specific HRP-conjugated secondary antibodies (1:3000 dilution, Thermo Scientific, Waltham, MA, USA). All blots were developed and quantified using ECL substrates (Millipore Inc., Billerica, MA, USA) and either a Gel Doc EZ Imager (Bio-Rad, Hercules, CA, USA,) or C-DiGit chemiluminescence Western blot scanner (LI-COR Biosciences, Lincoln, NE, USA).

For the analysis of rVSV particle protein content, 106 plaque forming units (PFU) of rVSV particles were mixed with the 5X SDS-PAGE sample buffer and boiled for 15 min at 93 °C. Subsequently, the proteins were separated via 8% SDS-PAGE gel and transferred to nitrocellulose membranes. Membranes were blocked in 1% bovine serum albumin (BSA) (in TBST containing 0.1% Triton X-100) for 1 h at room temperature, then incubated overnight at 4 °C with various antibodies: mouse anti-gp120 (B13) mAb, mouse anti-VSV-N (10G4) mAb (1:1000; Kerafast), mouse anti-Zaire Ebola virus glycoprotein (4F3) mAb (1:1000; IBT Bioservices), and rabbit anti-VSV-G (1:1000; Abcam). Membranes were then washed in 1X TBST and incubated for 1.5 h with the appropriate species-specific HRP-conjugated antibodies (1:1000 for all; Thermo Scientific). All blots were developed and quantified using ECL substrates (Millipore Inc., Billerica, MA, USA) and a Gel Doc EZ Imager (Bio-Rad).

### 2.8. Cell-to-Cell Fusion Assay

The fusion functionality of chimeric HIV-1 Env proteins was investigated using a cell–cell fusion reporter assay adapted from the HIV Veritrop assay [43]. In brief, HEK-293T cells were co-transfected with chimeric Env and pTat expression vectors. pREC.nfl.NL4-3 and pTat were transfected as positive and negative controls, respectively. Twenty-four hours post-transfection, HEK-293T cells were lifted with 6.25 mM EDTA and washed with complete DMEM. TZM-bl cells, an HIV Tat-inducible β-galactosidase reporter cell line, and were seeded together with transfected HEK-293T cells at 1.0 × 10^5^ cells/well of each cell type in a flat-bottom 96-well plate. Cells were then centrifuged at 400× *g* for 30 s to aid in cell adhesion. Twenty-four hours later, cell-cell fusion was analyzed using the Galacto-star assay kit (ThermoFisher Scientific). Briefly, media was removed, and cells were lysed for 10 min in Galacto-star lysis solution with agitation at 600 rpm. Cell debris was pelleted at 1700× *g* for 5 min and 10 µL of supernatant was transferred to a new white solid bottom 96 well plate. After the addition of 100 µL of Galacto-star reaction buffer and incubation at RT for 50 min, the light emission was measured with a BioTek Cytation 5 Imaging reader using the standard luminescence protocol on the Gen5 software (BioTek Instruments, Inc. Winooski, VT, USA). To further confirm chimeric Env-mediated cell–cell fusion, HEK-293T cells were pre-incubated with either HIV fusion inhibitors, maraviroc (MVC; 40 µM) or T-20 (10 µM), or the Anti-HIV-1 gp120 Monoclonal neutralizing Abs VRC01 and PGT121 for 1 h prior to combination with TZM-bl cells. Stock solutions of MVC and T-20 were, respectively, diluted in PBS and ethanol and then filter-sterilized. T-20 was obtained from the AIDS Research and Reference Reagent Program. VRC01 was utilized at various concentrations and obtained through the NIH AIDS Reagent Program, Division of AIDS, NIAID, NIH: Anti-HIV-1 gp120 Monoclonal (VRC01), from Dr. John Mascola (cat# 12033) [44].

### 2.9. HIV Pseudovirus Infectivity Assay

To further characterize the functionality of chimeric Env proteins designed in this study, HIV pseudoviruses expressing these chimeric Env were produced by co-transfecting HEK-293T cells with Env expression vectors and the S3G-ΔEnv plasmid at a 1:3 ratio (1 µg:3 µg for a 10 cm dish). After 48 h, pseudovirus-containing supernatants were collected, filtered through a 0.45 µm filter, and added to wells of a 96-well plate containing 2.0 × 10^4^ TZM-bl cells in 2-fold serial dilutions. Cell–pseudovirus combinations were incubated for 48 h at 37 °C to allow for the entry and expression of the HIV Tat inducible β-galactosidase. The virus infectivity was analyzed using the Galacto-star assay kit as described for the cell–cell fusion assay. As a positive control, HIV pseudoviruses expressing VSV-G were produced.

### 2.10. Statistical Analyses

GraphPad Prism v6.01 was used for all statistical analyses. *p* values and statistical tests are stated where appropriate. *p* values less than 0.05 re deemed significant.

### 2.11. Additional Methods

The Appendix A provides details on the more common methods used in this study, including cell transfections, Western blots, ELISAs, assessment of neutralizing antibodies, ELISPOT, and flow cytometry.

## 3. Results

### 3.1. Testing the Expression and Immunogenicity of HIV-1 Env in a VSV Vector

The studies by Parks et al. describe the protection of macaques from SHIV infection following vaccination with a VSVΔG vector expressing a chimera of Env.BG505 ectodomain (EC) with VSV G transmembrane (TM) and intracytoplasmic (CT) domains [45]. To obtain a sufficient vector for these vaccinations, the vaccine requires the propagation of the recombinant (r) VSV vector in CD4+/CCR5+ cell lines driven by the expression of the HIV-1 Env chimera, which proves to be difficult [21,28]. Furthermore, the low-level replication of the vaccine vector in vaccinated individuals would require replication in CD4+/CCR5+ cells (e.g., CD4+ T cells and macrophages), the same cell types susceptible to HIV-1 infection. Given the low-level in vitro propagation of the rVSV∆G_Env.BG505, we employed the approved rVSV-ZEBOV (Ervebor^TM^) to express different HIV-1 Env chimeric proteins [30]. With this vector, the EBOVZaire_glycoprotein (GP) drives the non-pathogenic propagation while the HIV-1 Env chimera is still expressed to elicit an HIV-specific immune response.

For vector optimization, rVSV vectors were first constructed to express the lab-adapted HIV-1 subtype B strain (NL4-3) Env containing the truncated CT (t713_Env), discovered to maintain high levels of incorporation in VSV particles [26]. We also cloned a chimeric HIV-1 NL4-3_Env with the TM and CT of SIV_mac239__Env (NL4-3_Env-EC/SIV_Env-TMCT) into either the rVSV with an intact VSV G (rVSV_G) or the VSV G substituted for the EBOV GP referred to as rVSV∆G+EBOV_GP in this study (for consistency with the nomenclature for all the constructs). However, it is important to note that the rVSV∆G+EBOV_GP is identical to the rVSV-ZEBOV (Ervebor^TM^), the FDA-approved Ebola vaccine. Following the rescue and propagation of both the rVSV_G and rVSV∆G+EBOV_GP vectors, western blot analyses reveal a low-level expression of the NL4-3t713_Env as compared with the chimeric HIV-1 NL4-3_Env-EC/SIV_Env-TMCT Env (Figure 1A–C) on the rVSV. The substitution of VSV_G with the EBOV_GP coding region in the rVSV vector results in a safe, effective, and approved EbolaZaire vaccine. As such, the rVSV∆G+EBOV_GP will serve as the vector to express the most immunogenic Env with the highest expression level in this study.

Using 6–8 week-old BALB/c mice, the immunogenicity of each of the four aforementioned rVSV-HIV vectors was determined. One intramuscular injection (IM) of one of the four rVSV vectors was administered to groups of seven mice and sera were collected four weeks post-immunization. None of the four rVSV vectors induce high levels of anti-HIV_Env binding antibodies, with OD 405 nM readings between 0.1 and 1.2 in undiluted sera (Figure 1D). As described below, the anti-Env-binding antibody levels increase over 200-fold following the IM immunization of the same mouse strain with an rVSV expressing an optimized Env chimera. Nonetheless, rVSV particles expressing EBOV_GP in place of VSV_G do have significantly higher anti-Env antibody responses for NL4-3t713_Env (*p* < 0.05, student’s *t*-test) (Figure 1D). There is a trend for similarly increased anti-Env antibody responses to the NL4-3_Env-EC/SIV_Env-TMCT in the rVSV∆G+EBOV_GP versus the rVSV_G vector (Figure 1D). Despite higher binding antibody responses to the chimeric NL4-3_Env-EC/SIV_Env-TMCT over the NL4-3t713_Env in the rVSV_G vector, this difference does not reach significance in the rVSV∆G+EBOV_GP vector. Furthermore, we do not detect the neutralization of the subtype A Q23 HIV-1 or the subtype B SF162 HIV-1 with sera from any of the vaccinated mice.

### 3.2. Generation and Design of Novel Chimeric HIV-1 Env Immunogens

Since the previous rVSV vectors express low levels of the chimeric NL4-3 Env and result in low anti-gp140 antibody titers, we attempted to optimize the Env chimera expression and immunogenicity. Eight novel chimeric HIV Env immunogens were screened for expression and function and then cloned into replication-competent rVSV vectors that could be propagated to high titers for future vaccination studies in mice and macaques. We surmised that the chimeric Env that proves functional for host cell entry and with the highest level of VSV expression may also be the best immunogen to elicit effective cellular and antibody immune responses.

To establish the best Env chimera, we first used a plasmid vector (pcDNA3.1) to determine if the expression could be increased and function could be maintained when the TMCT of HIV-1 NL4-3_Env (Figure 2A) is replaced with the TMCT of SIVmac239 Env or EbolaZaire GP (Figure 2B). As shown in western blots, the chimera containing the HIV-1 NL4-3_Env ectodomain (EC) with the SIV_Env-TMCT has higher levels of expression than observed with the HIV-1 NL4-3_Env-EC/EBOV_GP-TMCT (Figure 2C). In comparison with uncleaved, full-length 160 kDa HIV-1 Env (see control pREC-nfl lane in Figure 2E), the shorter TMCT of EBOV_GP and of SIV_Env (with a truncated CT) in this Env chimera results in an uncleaved Env glycoprotein of approximately 130–140 kDa (Figure 2C). After furin protease cleavage at the HIV-1 Env chimera’s “gp140-gp41”, the resulting EC (110–120 kDa) product is detected on the western blot (Figure 2C) for all NL4-3-based Env chimeras with the primary B13 antibody. Regardless of the chimeric Env construct, the EC product is only derived from the NL4-3 sequence (Figure 2A,B). An increased expression of chimeric NL4-3 Env glycoprotein involves co-transfection with the HIV-1 Rev (Figure 2C,F). Rev interacting with the RRE in the env coding sequence enhances nuclear export of this HIV-1 mRNA, leading to increased HIV-1 Env translation and expression.

As shown in Figure 1, the Env of the HIV-1 laboratory strain NL4-3 has low immunogenicity (Figure 1). To enhance immunogenicity, we constructed Env chimeras using the Env coding sequence from a primary subtype A HIV-1 isolate, A74 (Figure 2D). This A74 Env also harbors the K425 polymorphism shown to increase CD4 binding affinity and enhance immunogenicity [39,46]. The chimeras of HIV-1 A74_Env-EC with a TMCT from SIV Env_mac239T_ or from Ebola GP_Zaire_ are barely detected on the western blot (i.e., even lower expression than the equivalent NL4-3 Env chimeras) (Figure 2B,F). The inclusion of Rev provides only a modest increase in the expression of these A74 Env chimeras. HIV-1 A74_Env-EC/VSV_G-TMCT cannot be detected upon repeated transfections with or without Rev.

Previous studies describe low levels of expression of the subtype A BG505 Env chimera with the VSV_G-TMCT [28]. Codon optimization significantly increases the expression of their HIV-1BG505_Env-EC/VSV_G-TMCT. The codon adaptation index value (CAI) for the A74 env coding region is similar to that of the NL4-3 env coding region (0.69 versus 0.67) [42,47], suggesting that other factors may be responsible for the lower-level expression of the A74 versus NL4-3 chimeric Env (Figure 2C). With the codon optimization (CO) of the A74_Env-EC chimeras, there is a substantial increase in protein expression (Figure 2C,E,F) and without the “assistance” of Rev. Even the chimera of CO A74_Env-EC with the VSV_G-TMCT are now expressed to a similar level as the CO A74 Env-EC chimeras with TMCTs of EBOV_G and SIV_Env (Figure 2E,F).

The addition of Rev has no effect on the CO A74_Env chimeras (Figure 2E and Appendix A). This is not surprising, considering that the codon optimization of Env also mutates the RRE RNA sequence. Interestingly, Rev still does not enhance the expression of the codon-optimized chimera when the mutated RRE (due to CO changes) is replaced with the wild type RRE (Figure 2D and Appendix A). It is important to note that the RRE-Rev functions to increase transcript stability and transport, as well as to rescue HIV-1 mRNA from splicing. In the HIV-1 mRNA messages in which splicing sites are absent, enhancement of mRNA transport from the nucleus, mediated through RRE-Rev interaction, is related to reducing the impact of negative Cis-regulatory RNA elements (CREs) in the HIV-1 mRNA [48]. Codon optimization destroys nine of these putative negative CREs in the EnvA74-EC coding region/mRNA, suggesting this “codon-optimized” HIV-1 mRNA may be more related to destroying the CREs and possibly eliminating the need of Rev-RRE interactions. Only 3 CREs are predicted within the NL4-3_Env-EC coding region.

Relative to β-actin protein levels, all of the CO_EnvA74 chimeras are expressed at 5- to 7-fold higher levels than the wild type (non-codon optimized) forms of NL4-3_Env chimeras and over 10-fold higher levels than the wild type A74_Env chimeras.

### 3.3. Phenotype, Function, and Inhibition of the Chimeric A74 Env Glycoproteins

Previous studies reveal that an Env immunogen functional for receptor binding and host-cell entry may have a higher probability of inducing effective anti-HIV immune responses [46,49,50]. The surface expression of the Env chimeras is detected with the anti-B12 gp120 antibody on the majority of transfected HeLa cells (Figure 3A). However, the density of the Env chimeras per cell increases at least 100-fold with codon-optimized A74_Env-EC and regardless of whether the TMCT domain of SIV_Env, EBOV_GP, or VSV_G is used (Figure 3B).

Next, we determine if these codon-optimized Env chimeras expressed on the surface of effector HEK-293T cells (co-transfected with a Tat-expressing vector) could mediate fusion with target TZM-bl cells expressing CD4 and the CXCR4 and CCR5 coreceptors. Upon cell-to-cell fusion, the Tat expressed in the effector HEK-293T cells will activate the expression of β-Galactosidase driven by the HIV-1 LTR promoter in the target TZM-bl cells. High levels of cell fusion are mediated by the effector cell transfected with the pREC-nfl construct, which expresses all the HIV-1 proteins including HIV-1 NL4-3 Env [51,52]. Cell-to-cell fusion in this positive control is blocked via the fusion inhibitor Enfuvirtide (T20) that targets the EC domain of gp41. The CCR5 antagonist Maraviroc does not inhibit cell fusion mediated by the CXCR4-using NL4-3_Env. All of the CO A74_Env chimeras expressed on the effector HEK-293T cells (co-transfected with Tat) could mediate cell-to-cell fusion with the target TZM-bl cells. Cell fusion is inhibited by Maraviroc, which verifies the usage of CCR5 as the co-receptor for the A74 Env, even as a chimera. Surprisingly, none of the three CO A74_Env-EC chimeras with the TMCT of SIV_Env, EBOV_GP, or VSV_G re inhibited by Enfuvirtide, despite all having the “gp41” EC domain of HIV-1. As described below, the linkage of the HR1/HR2 domains of the HIV-1 Env to the TMCT of these SIV, EBOV, or VSV glycoproteins may have altered the Enfuvirtide binding site on HR1. Cell fusion mediated by the A74_Env chimeras is also sensitive to inhibition by the neutralizing antibodies PGT121 (binding to the V3 loop; Figure 3E), and VRC01 (binding to the CD4 binding site; Figure 3F). The relative inhibition by PGT121 and VRC01 of this cell fusion-mediated A74_Env chimera is nearly identical to the PGT121 and VRC01 inhibition of cell entry by HIV-1 with the A74 Env [46]. These findings suggest that the A74_Env-EC has similar sensitivity to these two neutralizing antibodies regardless of whether the TMCT is derived from HIV-Env, VSV_G, EBOV_GP, or SIV_Env. NL4-3 is inhibited by VRC01 but not by PGT121. As controls, neither PGT121 nor VRC01 antibodies can inhibit the VSV_G-mediated cell-to-cell fusion.

The HIV-1 Env chimeras were tested for the ability to pseudotype the lentiviral vector, SG3-ΔEnv HIV-1, and then to mediate TZM-bl cell entry. All three CO A74_Env chimeras are able to pseudotype HIV-1 for host cell entry but the chimeras with SIV_Env-TMCT mediate 3- to 5-fold greater entry efficiency (*p* < 0.01) than those chimeras with the TMCT of EBOV_GP or VSV_G (Figure 3D). The positive control involves pseudotyping the HIV-based lentiviral vector with VSV_G for efficient host cell entry (Figure 3D).

### 3.4. Incorporating the Optimized Chimeric A74 Env Proteins into rVSV

Based on the expression and functional analyses above, the A74_Env-EC with or without codon optimization and linked to either EBOV_GP-TMCT (constructs #1–4, Figure 4) or SIV_Env-TMCT (constructs #5–7) were cloned into the rVSV∆G+EBOV_GP. We also replaced the HIV-1 Env signal peptide sequence with the melittin signal peptide (Hb-SP) to test for maximal expression on the VSV particles (constructs #4 and 7, Figure 4). Finally, since the EBOV_GP and the HIV-1 Env chimera could compete for expression on the VSV vector surface, the EBOV_GP was codon-deoptimized to reduce the expression in favor of the increased expression of the CO A74_Env-EC/EBOV_GP-TMCT (construct #3). As described in Figure 5, our codon-deoptimized version 1 (CDO-1) remains inefficient, resulting in undetectable EBOV_GP, whereas CDO-2 EBOV_GP is still detectable in a western blot of cell lysates. Thus, CDO-2 EBOV_GP was cloned into the rVSV∆G+EBOV_GP in place of the wild type EBOV_GP (construct #3, Figure 4).

Within the rVSV∆G+EBOV_GP vector, the highest level of Env chimera expression is observed with the CO A74_Env linked to the SIV_Env-TMCT (Figure 6A,C). With repeat rVSV propagations, this difference is not significant among the codon-optimized A74_Env-EC with either the SIV_Env-TMCT or EBOV_GP-TMCT. However, the CO A74_Env chimeras remains significantly higher than the wild type A74_Env chimeras on the surface of the rVSV. The use of the Hb-SP over the Env SP appears to have minimal effect. In repeat propagations of the rVSVs, codon deoptimization consistently reduces EBOV_GP on the rVSV but does not alter the expression of the HIV-1 A74_Env chimeras (Figure 6A,B), suggesting that these two glycoproteins are not competing for occupancy on the rVSV surface. However, the propagation is slower and less efficient in Vero cells with the rVSV with CDO-2 EBOV_GP versus wild type EBOV_GP (i.e., the approved rVSV-ZEBOV vaccine).

### 3.5. Immunogenicity of the rVSV with the A74_Env Chimeras

Mice (six per group) were immunized by IM with rVSV∆G+EBOV_GP vectors expressing either wild type A74_Env-EC/SIV_Env-TMCT (construct #5, Figure 4), CO A74_Env-EC/SIV_Env-TMCT (construct #6), or CO A74_Env/EBOV_GP-TMCT (with the Hb-SP) (construct #4). Three additional groups were immunized by IM with the rVSV∆G+CDO-2_EBOV_GP vector expressing CO A74_Env-EC/EBOV_GP-TMCT (construct #3). For these three groups, the rVSV vaccine vector used for vaccination was purified by three different techniques: (i) resuspension of the pellet after ultracentrifugation, (ii) removal from a sucrose cushion after centrifugation, and (iii) retention after centrifugation on a 100 kDa size exclusion column. Sera and spleens were collected four weeks post-immunization. Splenocytes were incubated with a pool of 37 peptides (15-mers) derived from a gp120 subtype B consensus peptide pool and then assessed for the number of spot-forming units (SFU) expressing interferon gamma (IFN-γ). In the mice immunized with the rVSV vectors, low numbers of IFN-γ-producing, activated CD4+ and CD8+ T cells are detected upon incubations with Env peptides (Figure 7A). The dotted line shows the background SFU per million splenocytes. The peptides in the pool were selected based on the best conservation between consensus subtypes A and B in the C1 and V1 domain.

In mice immunized with rVSV expressing a codon-optimized version of the primary subtype A (A74) HIV-1 Env with the TMCT of SIV Env (constructs #3, 4, and 6, Figure 4), there are 200-fold higher levels of anti-gp140-binding antibodies than in mice immunized with the Env of laboratory strain NL4-3 or a chimera of NL4-3 with the TMCT of HIV-1 Env (constructs #II, III, V, and VI, Figure 1A). Through this re-iterative design process on the HIV-1 Env chimeras expressed on VSV vectors (described herein), there is an increase in Env-binding antibodies and HIV-neutralizing antibodies generated in immunized mice.

Typically, HIV-neutralizing antibodies (NAb) are a rare occurrence in the sera of mice immunized with an HIV vaccine. As noted earlier, no neutralization of HIV-1 was observed with the sera from mice immunized with the rVSV constructs expressing the truncated NL4-3 Env or NL4-3 Env chimera with the SIV_Env-TMCT (Figure 1D). In this experiment, at least 1 to 2 mice in every group immunized with the CO A74 Env chimeras result in sera with low-level neutralization of subtype A Q23 and subtype B SF162 viruses (HIV-1 pseudotyped with the Q23 or SF162 Env) (Figure 7C). However, only the group immunized with the rVSV expressing the CO A74_Env-EC/SIV_Env-TMCT has all six mice with neutralizing antibodies in their sera. Interestingly, mice 6-5 in this group show a high NAb titer of 50% (i.e., neutralization of Q23 and SF162 HIV-1 when sera re diluted over 300-fold) (Figure 7C). A similar NAb response is mounted with the rVSV constructs with the CO A74_Env-EC/EBOV_GP-TMCT, regardless of the presence of HP-SP versus wild type SP, the method of vector purification, or co-expression with a wild type versus codon-deoptimized EBOV_GP. Vaccination with all of these constructs/conditions results in lower NAb titers when compared with the sera obtained from mice immunized with CO A74_Env-EC/SIV_Env-TMCT. A lack of NAbs is observed in sera of mice immunized with the rVSV expressing the wild type (non-codon optimized) A74_Env-EC with the SIV_Env-TMCT. Sera from mock-vaccinated or untreated mice have a background inhibition of the Q23 and SF162 HIV-1 with a 50% neutralization at less than 1/40 sera dilution.

## 4. Discussion

We generated immunogenic, replication-competent rVSVΔG+EBOV_GP vaccine constructs that express high levels of novel, functional chimeric HIV-1 Env on rVSVΔG+EBOV_GP particles. Our chimeric HIV-1 Env immunogens were generated using a Subtype A HIV-1 Env and optimized through a reiterative process. Parks et al. have shown that replacing the TM and CT domains of HIV-1 Env with those of VSV-G produces a chimeric HIV-1 Env protein that is efficiently incorporated into rVSV particles in trimeric form (VSVΔG-Env.BG505) [28]. While showing promise, this chimeric Env.BG505 is unfortunately easily excluded from rVSV particles when co-expressed with VSV-G [21]. Without VSV-G, rVSV∆G only expresses the chimeric Env.BG505, which is difficult to propagate for vaccine production. Therefore, to generate Env chimeras with an enhanced stability and expression on rVSV particles, HIV-1 A74 Env chimeras were designed with the TM and CT domains of either SIV Env or EBOV GP. The best A74 Env chimera is co-expressed with EBOV_GP in a rVSV∆G, i.e., the approved rVSV-ZEBOV vector.

The ability of an HIV-1 Env immunogen to elicit protective anti-Env antibody responses requires a trimeric Env structure, preferably associated with a viral membrane that mimics the native HIV-1 Env structure [46,49,50,53]. Improved immunogenicity could also relate to an HIV-1 Env trimer binding its receptor/co-receptor and possibly exposing hidden, conserved Env epitopes for antigen presentation and immune recognition/response [49,50,54]. The function of the chimeric A74 Envs has to be analyzed independent of the rVSV-ZEBOV due to the co-expression of EBOV_GP. Ebola virus replicates in most cell types based on the entry mediated by EBOV-GP binding to membrane proteins with widespread cell expression, such as the NPC intracellular cholesterol transporter 1 (NCP-1) and/or T cell immunoglobulin mucin domain-1 (TIM-1) [55,56]. HEK-293T cells that express the chimeric Env’s could fuse with TZM-bl target cells expressing the HIV-1 receptors CD4 and CCR5. This cell-to-cell fusion is blocked by the addition of the CCR5 antagonist Maraviroc, confirming the binding interaction between chimeric Env and CCR5. Host-cell entry through CD4 and CCR5 is also mediated by a lentiviral HIV-1 ∆Env viral vector when pseudotyped with the A74 Env chimeras.

Despite the Enfuvirtide (T20) inhibition of cell-to-cell fusion mediated by NL4-3 Env, T20 can not inhibit fusion when mediated by the HIV-1 A74 Env chimeras. However, the virus with the full-length A74 Env is highly sensitive to T20 inhibition [39,46]. Based on several publications on the mechanisms of T20/Enfuvirtide inhibition and resistance, it is clear that minor changes in the HR1 and HR2 domains in gp41 can have profound effects on the T20 inhibition of the six alpha helix bundle formation, the ensuing pore formation, and membrane fusion leading to entry. We suspect that the linkage of the HIV-1 Env ectodomain to the TMCT of Ebola GP, VSV G, and SIV Env gp41 regions slightly alters the conformation and as such, prevents T20-binding. Six alpha helix bundle formation and membrane fusion still occurs with these chimeras. It is important to note that HIV-1 resistance to T20 inhibition is quite common in vitro and in vivo through mutations in the HR1 and HR2 domains without disrupting membrane fusion [57,58,59,60]. In many cases, this T20 resistance is linked to a reduction in HIV-1 fitness and reduced entry efficiency [59] but resulting in T20-resistant HIV-1 that is still capable of host-cell entry [57].

Previous studies have shown that the tier III/IV HIV-1 A74 is not inhibited by most broadly neutralizing antibodies (bNAbs) including the CD4 binding site antibodies VRC01 and N6 [46]. When the N425K polymorphism is introduced into the A74 Env, this virus becomes sensitive to CD4 binding site antibodies, including VRC01 and N6, while maintaining the same sensitivity to other classes of bNAbs, i.e., remaining sensitive to PGT121, which targets the V3 loop, but resistant to antibodies targeting the V1/V2 and MPER domains [46]. The A74 Env chimeras, all of which contain the N425K polymorphism, are sensitive to inhibition by VRC01 and PGT121. As in previous studies, the tier I laboratory strain NL4-3 Env is inhibited by VRC01 but insensitive to PGT121 inhibition. Despite the foreign TMCTs, the primary A74 virus and all of the derived A74 Env chimeras share the same sensitivity to inhibition by VRC01 and PGT121. These findings suggest the A74 gp120 region is targeted by these antibodies for inhibition, even when associated with the chimeric “gp41” (chimeric Env).

The CO A74 Env chimeras are found at similar levels on rVSV when Env translation is initiated with the wild type SP or the Hb-SP [16,61]. In the optimization process, we consider that the co-expression of EBOV_GP could reduce the levels of the A74_Env chimeras if there is competition for space on the rVSVΔG particle. EBOV_GP serves as the immunogen in the approved rVSV-ZEBOV vaccine [19,25,62]. Herein, we also observe an anti-GP antibody response in mice immunized with rVSV-ZEBOV co-expressing the HIV-1 A74 Env chimeras. In addition to serving as an immunogen on rVSV-ZEBOV, the EBOV_GP replaces VSV_G to facilitate high-titer replication necessary for vaccine production [29]. Upon vaccination with the rVSV-ZEBOV constructs, non-pathogenic replication of the vector enhances immunogenicity [30,62]. Most cell types support rVSV-ZEBOV replication through EBOV-GP entry, but only a small subset of immune cells can support an rVSV∆G with only the HIV-1 Env, as is the case with the VSVΔG-Env.BG505 construct [28]. To maintain the balance of vector propagation through EBOV_GP while maximizing chimeric A74 Env expression to elicit an immune response, we codon-deoptimize the EBOV-GP within the rVSV-ZEBOV co-expressing the A74 Env chimeras. Despite a ~2.5-fold decrease of EBOV_GP (per rVSV particle) with codon deoptimization, the expression level of the CO A74 chimera does not increase (or decrease) on the rVSV particle. These findings suggest that the HIV-1 Env and the EBOV Env are not in competition for space on the rVSV membrane.

We have observed antibody responses to both the HIV-1 and EBOV envelope glycoproteins on the rVSV vector surface. Immune responses to the EBOV GP and HIV-1 Env in the rVSV-ZEBOV may be advantageous as a dual-protective vaccine. However, unlike a single immunization that can provide protection against Ebola virus, preliminary studies suggest that a prime-boost may be required for an HIV-1 preventative vaccine by this rVSV vector. Anti-EBOV_GP responses with a prime immunization may reduce the effectiveness of the boost. Recent studies with a rVSV expressing the VSV G co-expressing chimeric SARS-CoV-2 S proteins result in vector-based immune responses and yet, prime-boost immunization with this VSV-based SARS-CoV-2 vaccine has still shown some of the highest neutralizing antibody levels to date [63]. Regardless, we may still require determination of the correct balance of EBOV GP and HIV-1 Env chimeras on the rVSV vector for the most effective anti-HIV-1 Env immune responses with the boost.

This study is designed to generate and select the optimal HIV-1 Env chimera for maximal, functional expression and immunogenicity. This involves testing 15 different HIV-1 Env chimeras for optimal expression on rVSV with or without EBOV_GP, immunizing over 60 mice with ten of these vectors, and then measuring the anti-gp120 responses, neutralizing antibodies, and T cell responses. Nonetheless, these preliminary immune analyses re performed in mice, which is not a suitable model to support future human studies. The rVSV∆G+EBOV_GP vector expressing the codon optimized version of A74_Env-EC/SIV_Env-TMCT is now being used to vaccinate nonhuman primates (macaques) for more in-depth analyses of host immune responses and correlate these anti-HIV-1 Env responses to protection from virus challenges.

## 5. Conclusions

We conclude that an HIV-1 Env with even a truncated TMCT is still expressed on the rVSV vector surface at much lower levels than an HIV-1 Env chimera with the TMCT of SIVmac239 Env. However, this Env chimera derived from the NL4-3 laboratory strain placed into a rVSV vector shows poor immunogenicity in vaccinated mice. The subsequent reiterative modification of a primary A74 Env maintains the chimeric Env function and increases the expression on the rVSV surface, regardless of whether the TMCT is derived from VSV, SIV, or EBOV. Following mouse immunizations with the optimized Env chimeras in rVSV-ZEBOV, we observe similar levels of anti-gp140 antibodies and cell-mediated responses to Env peptides, again, regardless of which CO A74 chimera is employed (with TMCT from EBOV_GP or SIV_Env). Nonetheless, mice vaccinated with a VSV vector expressing the CO A74_Env with the SIV_Env-TMCT have sera capable of neutralizing a subtype A and B HIV-1 strain. In contrast, the same VSV vector expressing the CO A74_Env with the EBOV_GP-TMCT induces significantly lower levels of neutralizing antibodies. Despite similar expression levels of these Env chimeras on the VSV particles, there is the possibility that the HIV-1/SIVmac239 Env chimera on VSV is just a better approximation of the native Env gp120/gp41 on the HIV-1 virus. The rVSV-ZEBOV with the codon-optimized Env ectodomain of A74 and the TMCT of SIVmac239_Env is now the vaccine selected for future studies.

## Figures and Tables

**Figure 1 vaccines-11-00977-f001:**
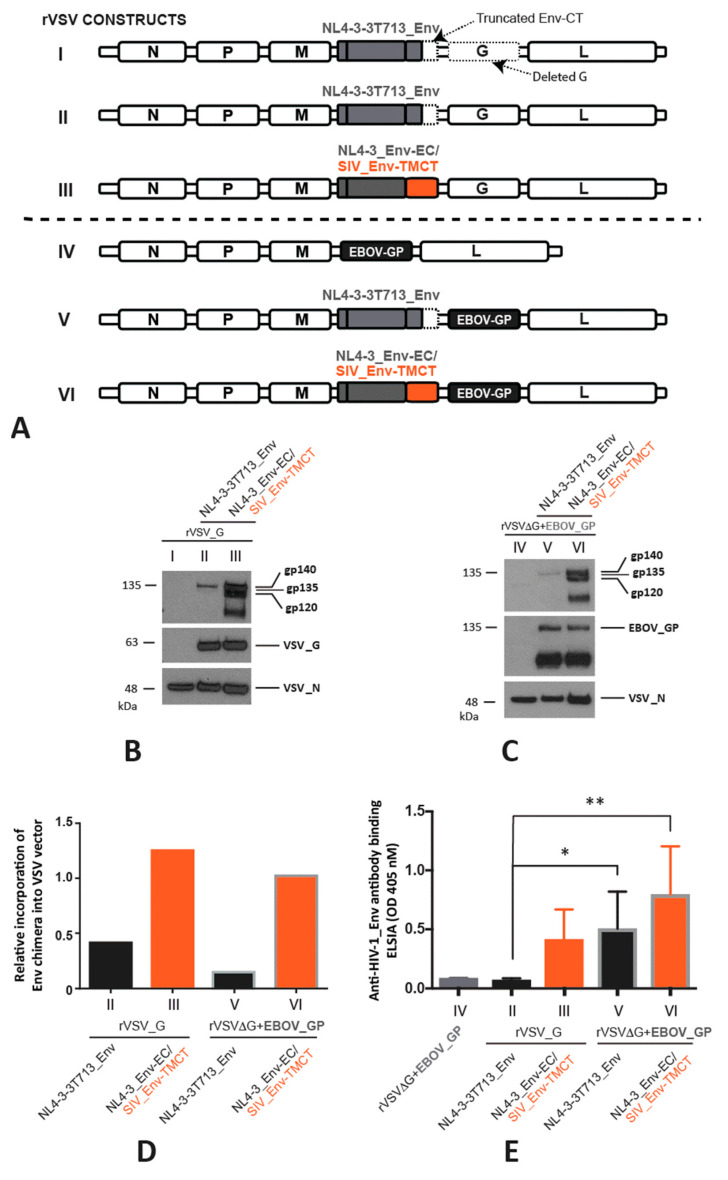
Expression and immunogenicity of HIV-1 Env on VSV vectors. (**A**) The HIV-1 Env with a truncated C-terminal cytoplasmic tail (CT) (NL4-3-3T713_Env) and chimeric HIV-1 Env with an ecto/extracellular domain of gp120/gp41 and the transmembrane (TM) and truncated CT of SIV Env, shown in orange, (NL4-3_Env-EC/SIV_Env-TMCT) were expressed when cloned in cis into a recombinant (r) VSV vector (first two constructs I and II) or into an rVSV in which the VSV G glycoprotein gene was deleted and replaced with the Ebola virus glycoprotein gene (EBOV_GP) (next three constructs, III, IV, and V). All HIV-1 Env sequences were derived from the laboratory strain NL4-3. Construct III is a control rVSV construct without the modified NL4-3 Env’s but with EBOV_GP replacing the VSV_G. The Western blots in (**B**,**C**) show the expression of the truncated NL4-3-3T713_Env (gp135 or 135 KDa) and of the NL4-3_EnviEC with the SIV_Env-TMCT (~gp140) using the primary anti-gp120 B13 antibody. The NL4-3 Env-EC is cleaved from the SIV_Env-TMCT, resulting in a gp120 detected by the B13 antibody. Blots were also probed to detect the VSV_G and VSV_N proteins (**B**) and EBOV GP and VSV_N proteins (**C**). The expressions of the various HIV-1 Env products relative to VSV_N are shown in (**D**) with the constructs with the SIV_Env-TMCT shown with an orange bar. Seven BALB/c mice in each of five groups were immunized by intramuscular injections with one of five rVSV constructs illustrated in (**A**) with the mice immunized with the constructs containing the SIV_Env-TMCT shown with an orange bar. Anti-HIV-1_Env antibody response in mouse sera was measured at 28 days post-immunization using ELISA, * = *p* < 0.05, ** = *p* <0.01 (**E**).

**Figure 2 vaccines-11-00977-f002:**
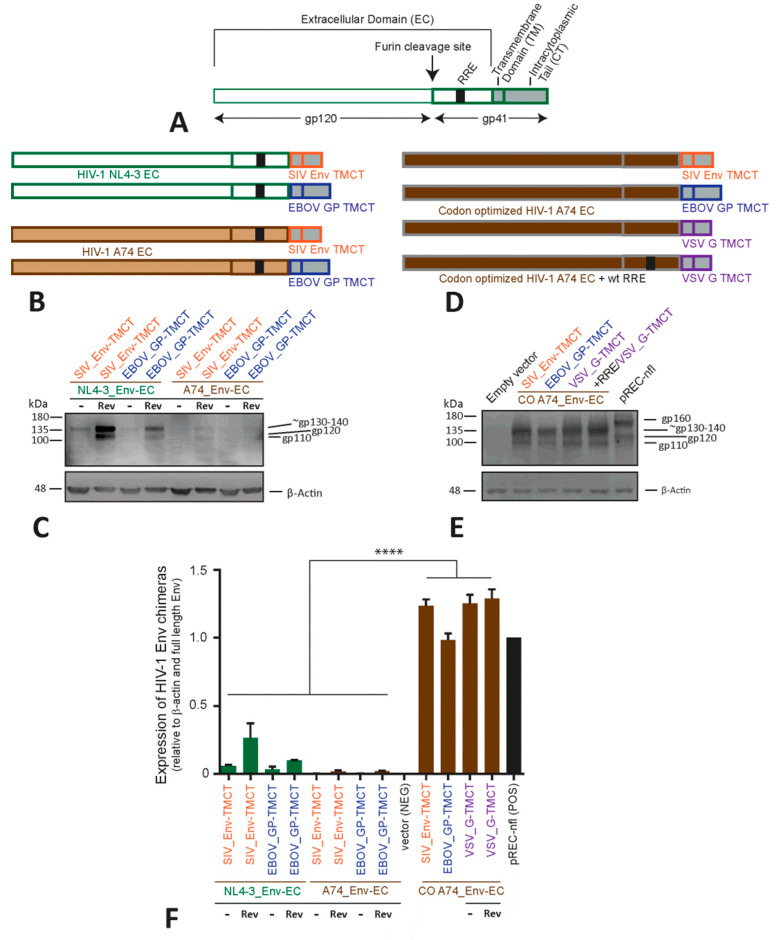
Generation and characterization of chimeric HIV-1 Env immunogens. (**A**) Schematic representation of a linear wild type (WT) HIV Env made up of two main subunits, gp120 and gp41. Gp120 and gp41 contain the Env signal peptide (SP), extracellular domain (EC), Rev-response element (RRE) (within the mRNA), transmembrane domain (TM), and cytoplasmic tail (CT). (**B**) Various Env chimeras were generated utilizing either HIV-1 NL4-3 (green) or A74 Env EC (light brown), and then replacing the HIV Env TM and CT domains with those of SIV Env (red) or EBOV glycoprotein (blue). The genes encoding these Env chimeras (**B**) were cloned into pcDNA3.1, transfected into HEK-293T cells, and the following Env protein expression was measured in cell lysates with Western blots for the anti-HIV-1 Env B13 antibody and probing with anti-β actin antibody as a loading control. (**C**,**D**) Similar Env chimeras were cloned in the pcDNA3.1 as described in (**B**) but now with a codon-optimized (CO) HIV-1 A74 EC (dark brown) linked to the SIV_Env (red), EBOV_GP (blue), or VSV_G (purple) TMCT. The expression of these chimeras in transfected HEK-293T cells is shown in the western blot (**E**). The expression of all the Env chimeras relative to β-actin (as measured by western blot (**C**,**D**)) is shown with **** representing *p* < 0.0001 (ANOVA) (**F**). The western blots in panels (**B**,**D**) were run on the same day and represent the data from transfections of all the constructs performed on the same day. To ensure the CO A74-EC bands were visible in panel (**E**), the exposure time is reduced as compared with that in panel (**D**).

**Figure 3 vaccines-11-00977-f003:**
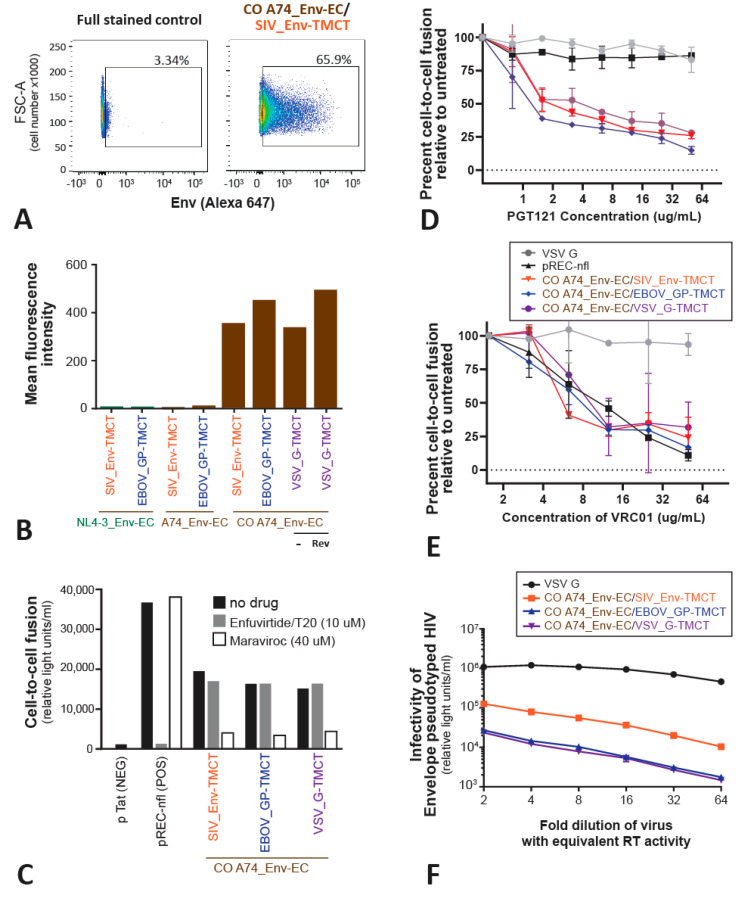
Surface expression and functional analyses of the Env chimeric proteins. Flow cytometry was performed on the Env chimeric proteins expressed on the surface of HEK293T cells following transfection with the pcDNA3.1 constructs (as described in the Materials and Methods) to measure the percentage of cells expressing the Env chimeras (as an example, CO A74_Env-EC/SIV_Env-TMCT surface expression in (**A**)). Deep red to dark blue represent a high to low number of cells. Throughout the figure, the color represents the CO A74_Env-EC with SIV_Env-TMCT (orange), EBOV_GP-TMCT (blue), and VSV_G-TMCT (purple). The mean level of expression per cell surface for all Env chimeras is shown in (**B**). The HEK-293T cells transfected with the various chimeras were mixed with TZM-bl cells to measure cell fusion mediated by the Env chimeras’ interactions with CD4 and CCR5 (or CXCR4) on TZM-bl cells (**C**). The level of cell fusion is measured by HIV-1 Tat, diffusing from the effector HEK-293T cells into the TZM-bl cells to activate LTR-driven expression of β-Galactosidase. The entry inhibitors Maraviroc (40 uM) and Enfuvirtide/T20 (10 uM) were added to the cell mixtures to block cell fusion by preventing Env binding to CCR5 or blocking gp41-mediated membrane fusion, respectively. The same cell-to-cell fusion assays were also performed (in triplicate) with increasing concentrations of anti-HIV broadly neutralizing antibodies PGT121 (**D**) and VRC01 (**E**). Finally, the pcDNA3.1 was transfected with an HIV-1∆Env expression vector to produce HIV-1 pseudotyped with the various Env chimeras. These pseudotyped viruses were serially diluted and used to infect TZM-bl cells, again measuring Tat-induced β-Galactosidase activity (**F**).

**Figure 4 vaccines-11-00977-f004:**
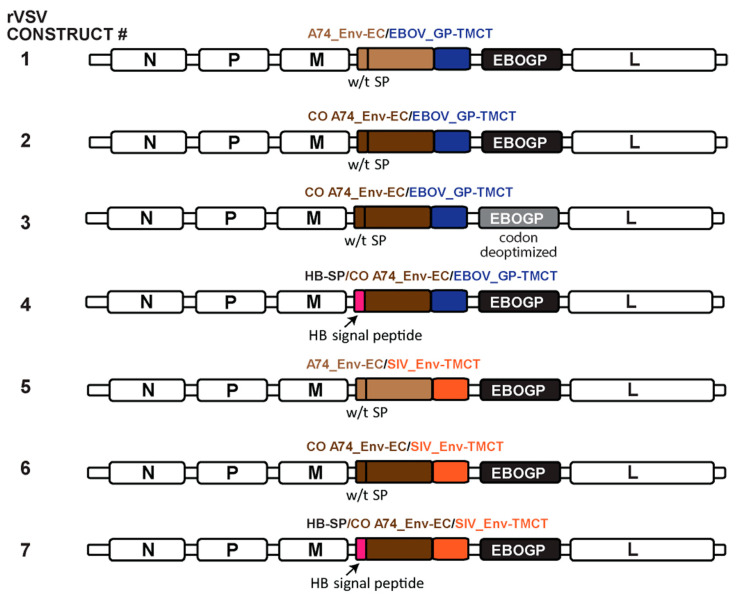
Schematic of rVSV genomes containing the various Env chimeras. The expression and function of the various Env chimeras described in Figure 2 were cloned with the EBOV_GP into the VSV_G gene position, in place of VSV_G. The only variation on the chimeric Env constructs described in Figure 2 was the inclusion of the honey-bee melittin signal peptide (HB-SP) in place of the Env SP in constructs #4 and #7. The EBOV_GP in construct #3 was codon-deoptimized. The color of the text represents the specific SP, EC, and TMCT inserts in the schematic representation.

**Figure 5 vaccines-11-00977-f005:**
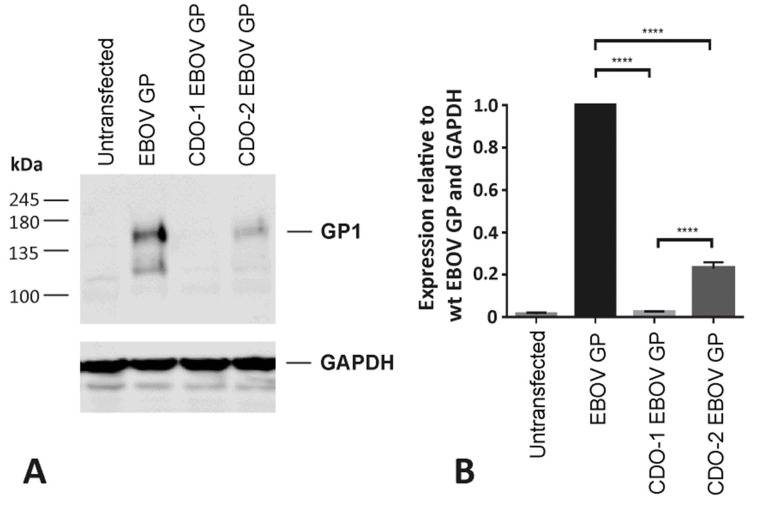
Generation of a codon deoptimized EBOV GP gene. (**A**) Two candidate EBOV GP genes, CDO-1 and CDO-2, were designed with the addition of rare codons and various negative cis-regulatory elements. Candidate EBOV GP genes were synthesized and cloned into the pcDNA3.1 vector via GenScript. HEK-293T cells were transfected with the EBOV GP pcDNA3.1 expression vectors. Forty-eight hours post-transfection, cell lysates were collected and analyzed for EBOV GP (GP1) expression via western blot. Samples were normalized to levels of GAPDH. (**B**) Densitometric analysis depicts EBOV GP expression relative to wild type EBOV GP (N = 3). One-way ANOVA, followed by Tukey’s multiple comparisons test, was utilized to determine statistical significance (**** = *p* < 0.0001).

**Figure 6 vaccines-11-00977-f006:**
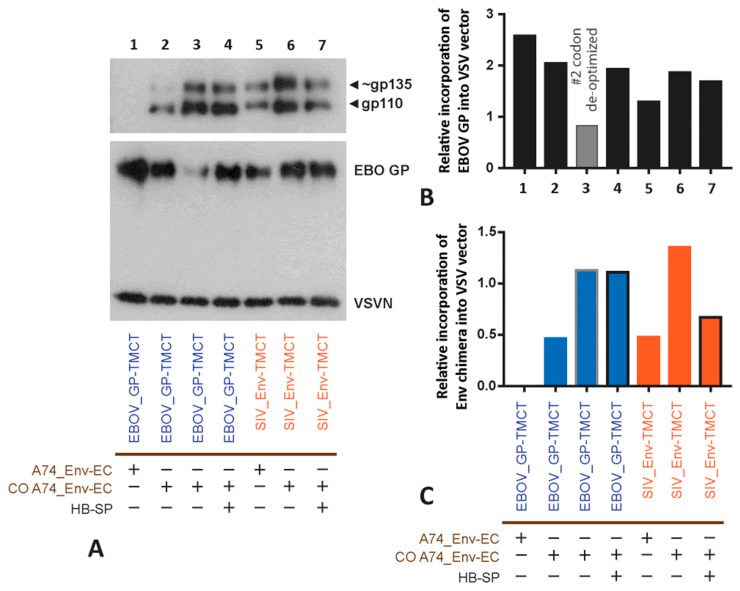
Examining the expression of primary A74 HIV-1 Env chimera on rVSV particles. (**A**) Purified rVSV particles were lysed and analyzed by western blot for chimeric Env and VSV-N content. VSV-N was used as a loading control. Incorporation of the EBOV_GP (**B**) and HIV-1 Env chimera (**C**) into rVSV was measured relative to the VSV_N expression following densitometry scanning. The colors represent the different constructs, i.e., the A74 and CO A74 with the EBOV_GP-TMCT (blue) or the SIV_Env-TMCT (orange).

**Figure 7 vaccines-11-00977-f007:**
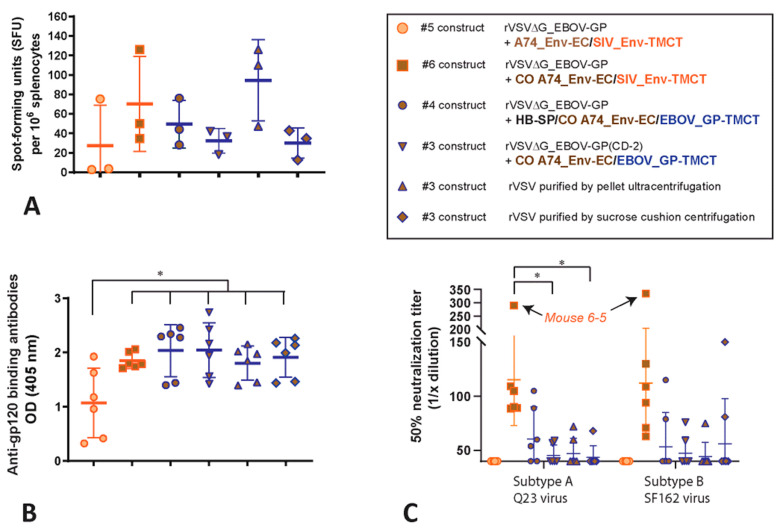
Immunogenicity of the HIV-1 A74 Env chimera with the rVSV constructs. Six mice in each of six groups were immunized by IM with the rVSV constructs #5, 6, 4 and with #3 concentrated/purified by retention after centrifugation on a 100 kDa size exclusion column (standard purification for all constructs), after resuspension of a pellet after ultracentrifugation, or removal from a sucrose cushion after centrifugation. After 28 days, (**A**) the splenocytes were isolated and incubated with Env C1-V1 peptides to determine the number of cells releasing IFN-γ (spot forming units) by ELISPOT. The sera after 28 days were measured for HIV-1 gp120 binding antibodies by ELISA (**B**) and for 50% neutralization of pseudotyped virus expressing the Tier 1, subtype A Q23 or Tier 1, subtype B SF162 Env using TZM-bl cells (**C**). * represents *p* < 0.05 (two-tailed T tests).

## Data Availability

The datasets used and/or analyzed during the current study are available from the corresponding author on reasonable request.

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
