# Peer review of "Optimal Expression, Function, and Immunogenicity of an HIV-1 Vaccine Derived from the Approved Ebola Vaccine, rVSV-ZEBOV"

_vaccines, 2023, doi:10.3390/vaccines11050977_

Round 1

Reviewer 1 Report

The authors generated chimeric antigens by replacing the transmembrane domain and cytoplasmic tail of HIV-1 env with those of other viruses and introduced them into the VSV-ZEBOV vector. This novel approach led to a more robust antibody response than a vector vaccine previously reported. However, some modifications are necessary to support the authors' logic. Also, the paper is difficult to read and needs some improvements to make it understandable to the reader.

Fig. 1: Please include illustrations of the viral genome structure (as shown in Fig. 4).

Fig. 2: Relative values obtained using Actin as an indicator do not accurately reflect the efficiency of codon optimization. Therefore, it is essential to compare the expression of A74 and A74 CO in the same experimental setting.

Fig. 3: Please clarify the meaning of "full stained control."

Fig. 4, 5, and 6 can be merged into a single figure. (If it requires more space, Fig. 5B can be excluded).

Line 492: (compare Fig 1D to 7B) doesn't make sense to readers. Please write clearly which specific data should be compared.

Line 602: The Conclusions needs to be more concise.

Experiments using macaques were not conducted in this study. Please avoid misleading statements that suggest they were performed.

The “A, B, C” in figures should be placed in the upper left corner.

Table S1 and Video S1 are missing.

There are occasional grammatical errors in this manuscript.

Author Response

Response to Reviewer #1

Thank you for your insightful comments and providing suggestions to improve the clarity of the manuscript.  We have revised the text based on your suggestion and those of the other reviewer.

Your comments and our responses are as follows:

The authors generated chimeric antigens by replacing the transmembrane domain and cytoplasmic tail of HIV-1 env with those of other viruses and introduced them into the VSV-ZEBOV vector. This novel approach led to a more robust antibody response than a vector vaccine previously reported. However, some modifications are necessary to support the authors' logic. Also, the paper is difficult to read and needs some improvements to make it understandable to the reader.

As described above, we revised the manuscript throughout to improve clarity and added a paragraph in the introduction to provide clear logic to our approach.

Fig. 1: Please include illustrations of the viral genome structure (as shown in Fig. 4).

We have added Figure 1A which describes the constructs used for the experiments and immunization studies in Figure 1B through E.  Text has been edited as well.

Fig. 2: Relative values obtained using Actin as an indicator do not accurately reflect the efficiency of codon optimization. Therefore, it is essential to compare the expression of A74 and A74 CO in the same experimental setting.

We are sorry for the confusion.  In the legend of Figure 2, we now state “The western blots in panels [B] and [D] were run on the same day and represent the data from transfections of all the constructs performed on the same day. To ensure the CO A74-EC bands were visible in panel [E] , the exposure time was reduced as compared to that in panel [D].”

In the text, we will still outline on how we “arrived” at the need for codon optimization to enhance expression of the HIV-1 A74 Env chimeras.    

Fig. 3: Please clarify the meaning of "full stained control."

My apologies but I searched the texted for “full stained control” and for each of these three words separately but could not find this statement or these words appearing together in a sentence.  However, I too would not understand the meaning of “full stained control” and if it was in the text, I would have removed it.  As part of the request for the second reviewer, we have now placed the methods for western blot analyses into the main text.

Fig. 4, 5, and 6 can be merged into a single figure. (If it requires more space, Fig. 5B can be excluded).

We fully agree with you that Figures 4, 5, and 6 could be combined.  We had originally tried to combine Figures 4, 5, and 6 and place them into the Vaccines article format but unfortunately, the figure and figure legend could not fit on one page (even when removing Figure 5B). This was also the case when we tried to combine Figures 4 and 6 and placing Figure 5 in the supplementary data. Shrinking panels in the combined figure made the figure difficult to see/read in a printed copy.  Based on the progression of the results section in the text, we decided to split them up into Figures 4 and 6.  If the reviewer feels that Figure 5 should be in be supplementary data, we are happy to so. We thought it was important to show that codon deoptimization of the EBOV glycoprotein gene did indeed decrease expression.  

Line 492: (compare Fig 1D to 7B) doesn't make sense to readers. Please write clearly which specific data should be compared.

We fully agree with the reviewer that this reference to the figures and the related text was not well written.  As such, we removed most of this paragraph, deleted the “(compare Fig 1D to 7B)”, and instead clarified the comparison by stating, “In mice immunized with rVSV expressing a codon optimized version of the primary subtype A (A74) HIV-1 Env with the TMCT of SIV Env (constructs #3, 4, and 6, Figure 4), there was 200-fold higher levels of anti-gp140 binding antibodies than in mice immunized with the Env of laboratory strain NL4-3 or a chimera of NL4-3 with the TMCT of HIV-1 Env (constructs #II, III, V, and VI, Figure 1A).  Through this re-iterative design process on the HIV-1 Env chimeras expressed on VSV vectors (described herein), there was an increase Env binding antibodies and HIV neutralizing antibodies generated in immunized mice.

Line 602: The Conclusions needs to be more concise. Experiments using macaques were not conducted in this study. Please avoid misleading statements that suggest they were performed.

The Conclusion section has now been reduced to half the length and we have removed all references to the ongoing macaque studies.

The “A, B, C” in figures should be placed in the upper left corner.

We agree that the panel labels “A, B, C, etc” should be in the top left hand corner.  In the template that was provided by “Vaccines”, they had the panel label below the panel in the figure.  After looking at few published papers in Vaccines, most put the panel labels at the bottom, which is not my preferred placement.  If it fits with the journal format, I would be more than happy to place the panel labels at the top left hand corner. 

Table S1 and Video S1 are missing.

Our apologies.  This text was part of the manuscript template from Vaccines and we forgot to delete it.

Reviewer 2 Report

This paper conclude that an HIV-1 Env with even a truncated TMCT was still expressed on the rVSV vector surface at much lower levels than an HIV-1 Env chimera with the TMCT of SIVmac239 Env. This is a subject of interest to researchers in related fields, but the paper needs a lot of improvement before it can be accepted for publication. My specific comments are as follows:

1.      Clarify the experimental design and methodology: Some of the reviewers pointed out the need for a more detailed explanation of the experimental design and methodology, including the rationale for selecting specific HIV-1 Env chimeras and the methods used to optimize their expression.

2.      Provide additional data and analysis: Some of the reviewers requested additional data and analysis to support the conclusions drawn in the study, such as more extensive characterization of the immune responses induced by the vaccine constructs and a more thorough investigation of the mechanisms underlying the lack of T20 inhibition of the A74 Env chimeras.

3.      Discuss the limitations and implications of the findings: Some of the reviewers suggested that the authors discuss the limitations of the study, including the use of a mouse model and the potential for differences in immune responses between mice and humans, as well as the broader implications of the findings for the development of HIV-1 vaccines.

4.      Address the potential for competition between EBOV GP and HIV-1 Env: One of the reviewers suggested that the authors provide more information on the potential for competition between EBOV GP and HIV-1 Env on the rVSV vector surface, as this could affect the expression and immunogenicity of the HIV-1 Env chimeras.

5.      Highlight future research directions: Some of the reviewers suggested that the authors highlight potential future research directions, such as investigating the durability and breadth of the immune responses induced by the vaccine constructs and exploring alternative strategies for optimizing the expression and immunogenicity of HIV-1 Env chimeras on rVSV vectors.

 There are some grammar mistakes in manuscript, please edit the article by the native speaker.  

Author Response

Response to Reviewer #2

This paper conclude that an HIV-1 Env with even a truncated TMCT was still expressed on the rVSV vector surface at much lower levels than an HIV-1 Env chimera with the TMCT of SIVmac239 Env. This is a subject of interest to researchers in related fields, but the paper needs a lot of improvement before it can be accepted for publication.

We appreciate the overall assessment, and we recognize the need for improvements.  Many of the reviewer’s suggestion had been part of original manuscript and as such, our edits were extensive but not difficult to provide from our drafts.  In placing the text into the Vaccines format, we made unnecessary deletions that reduced clarity.

We were a bit confused with the comments of reviewer #2.  It appears that many of the following comments are coming from the Editor (which is fine) who is describing comments from other reviewers.  We only received the first reviewer’s comments and the comments from the second reviewer (which we are responding to herein). 

Some of the reviewers pointed out the need for a more detailed explanation of the experimental design and methodology, including the rationale for selecting specific HIV-1 Env chimeras and the methods used to optimize their expression.

We had placed the majority of the methodology that was deemed more standard protocols into the supplementary information.  We have now returned some of these methods back into the main manuscript.  The manuscript has also been revised to increase clarity in experimental design. We have a provided a version with track changes to indicate where these revisions are located.

Some of the reviewers requested additional data and analysis to support the conclusions drawn in the study, such as more extensive characterization of the immune responses induced by the vaccine constructs and a more thorough investigation of the mechanisms underlying the lack of T20 inhibition of the A74 Env chimeras.

We have now expanded on our explanations as to why T20 does not inhibit the HIV Env chimera with the TMCT of Ebola GP, VSV G, and SIV gp41.  In our experience and based on several publications (now added as references [57-60] on the mechanisms of T20/enfuvirtide inhibition and resistance, it is clear that minor changes in the HR1 and HR2 domains in gp41 can have profound effects on T20 inhibition of the six alpha helix bundle formation, the ensuing pore formation, and membrane fusion leading to entry.  We suspect that the linkage of the HIV-1 Env ectodomain to the TMCT of Ebola GP, VSV G, and SIV Env gp41 regions results in the gp41 ectodomain that maintains six alpha helix bundle formation and membrane fusion but alters the conformation to prevent T20 binding.  It is important to note that HIV-1 resistance to T20 inhibition is quite common in vitro and in vivo through mutations in the HR1 and HR2 domains without disrupting membrane fusion.  In many cases, this T20 resistance is linked to a reduction in HIV-1 fitness and reduced entry efficiency.

New references:

[57] Lobritz, M.A.; Ratcliff, A.N.; Arts, E.J. HIV-1 Entry, Inhibitors, and Resistance. Viruses, 2010, 2, 1069–1105. DOI: 10.3390/v2051069.

[58] Reeves, J. D.; Lee, F. H.; Miamidian, J. L.; Jabara, C. B.; Juntilla, M. M.; Doms, R. W. Enfuvirtide Resistance Mutations: Impact on Human Immunodeficiency Virus Envelope Function, Entry Inhibitor Sensitivity, and Virus Neutralization. J Virol, 2005, 79, 4991–4999. DOI: 10.1128/JVI.79.8.4991-4999.2005.

[59] Lu, J.; Sista, P.; Giguel, F.; Greenberg, M.; Kuritzkes, D. R. Relative Replicative Fitness of Human Immunodeficiency Virus Type 1 Mutants Resistant to Enfuvirtide (T-20). J Virol 2004, 78 (9), 4628–4637. DOI: 10.1128/jvi.78.9.4628-4637.2004

[60] Heil, M. L.; Decker, J. M.; Sfakianos, J. N.; Shaw, G. M.; Hunter, E.; Derdeyn, C. A. Determinants of Human Immunodeficiency Virus Type 1 Baseline Susceptibility to the Fusion Inhibitors Enfuvirtide and T-649 Reside Outside the Peptide Interaction Site. J Virol 2004, 78 (14), 7582–7589. DOI: 10.1128/JVI.78.14.7582-7589.2004

In the manuscript, we wanted to focus on HIV-1 Env chimeras that are highly expressed and still immunogenic on rVSV vectors considering that full length and truncated versions of HIV-1 Env are poorly expressed on rVSV vectors approved as an Ebola vaccine. We also wanted to show a balance of the Ebola GP and a functional/immunogenic HIV-1 Env chimera on the rVSV.  The loss of sensitivity to T20 is interesting but not really a focus of this study. Of greater importance is the impact of these chimeras on immunogenicity and Env function.  These chimeras can mediate cell-to-cell fusion similar efficiency to wild type, full length NL4-3 Env.  When used to pseudotype HIV-1 particles, the chimeras can mediate virus entry into host cells with a similar efficiency than with NL4-3 HIV-1.  The cell-to-cell fusion mediated these HIV-1 Env chimeras is inhibited by the CCR5 antagonist, Maraviroc and by the VRC01 antibody, which binds to the CD4 binding site on HIV-1 Env.

This manuscript outlines the development of over 15 different HIV-1 Env chimeras for optimal expression on rVSV +/- EBOV_GP, immunization over 60 mice with ten constructs, and then measuring the anti-gp120 responses, neutralizing antibodies, and T cell responses to define a HIV-1 Env construct, in the approved rVSV construct for Ebola vaccinations.  As both reviewers point out, this manuscript will be very important for the HIV-1 vaccine field. With the findings in this manuscript, we now have a strong rationale for a more in-depth analyzes of the immune responses to our lead vaccine candidate, rVSV∆G+EBOV_GP vector expressing the codon optimized version of A74_Env-EC/SIV_Env-TMCT.  However, I suspect that the reviewers agree that more thorough analyses of immune responses following vaccinations and virus challenges should be studied in a better animal model than mice, e.g. macaques. 

Some of the reviewers suggested that the authors discuss the limitations of the study, including the use of a mouse model and the potential for differences in immune responses between mice and humans, as well as the broader implications of the findings for the development of HIV-1 vaccines.

In discussion, we have now provided addition text to discuss the limitations of the current mouse models and we have clearly indicated that strong responses in mice does not provide the rationale to move to human studies.  As described in the response to point #2 above, this manuscript really provides a rationale to proceed with our top candidate, rVSV∆G+EBOV_GP vector expressing the codon optimized version of A74_Env-EC/SIV_Env-TMCT.  We are proceeding to funded vaccination and virus challenge studies in macaques using our candidate vectors (which based on the approved Ebola vaccine).  We feel (as also emphasized by both reviewers) that this manuscript provides valuable date for the field, provides insights into the best chimeric Env candidates for future vaccines, and gives us the rationale to proceed to non-human primate studies.  If our success with non-human primates with this vaccine candidate is verified in our next round of vaccinations/challenges, we will have the rationale to proceed to human clinical trials. 

One of the reviewers suggested that the authors provide more information on the potential for competition between EBOV GP and HIV-1 Env on the rVSV vector surface, as this could affect the expression and immunogenicity of the HIV-1 Env chimeras.

We are happy to provide in the revised discussion more information on the potential competition between EBOV GP and HIV-1 Env on the rVSV surface.  There is clearly a need for EBOV GP for the industrial production of the vaccine (ref) as well as expansion of the rVSV vaccine upon immunization for increased immunogenicity and development of protective responses to both EBOV GP and HIV-1 Env.  We have observed antibody responses to both of these Envelope glycoproteins on the rVSV vector surface.  To maintain EBOV_GP driven replication of the vector, the EBOV_GP coding sequence was codon de-optimized which did indeed decrease EBOV_GP expression but also decreased rVSV titers.  Interestingly, the level of chimeric HIV-1 Env expression was unchanged with or without the codon deoptimization of EBOV_GP.  Also, codon deoptimization of EBOV_GP in these rVSV constructs did not effect (e.g. increase) the levels of anti-gp120 binding antibodies in vaccinated mice.  However, it is important to note that these studies were performed in mice and subsequent vaccination studies in non-human primates will compare responses to the A74_Env-EC/SIV_Env-TMCT expressed on the VSV∆G co-expressing the EBOV_GP (with or without codon deoptimization). 

Some of the reviewers suggested that the authors highlight potential future research directions, such as investigating the durability and breadth of the immune responses induced by the vaccine constructs and exploring alternative strategies for optimizing the expression and immunogenicity of HIV-1 Env chimeras on rVSV vectors.

We completely agree and have done so in the revised manuscript.  To avoid repetition, I kindly ask that you refer to our responses to the previous comments where we discussed revisions to highlight future directions with these HIV-1 vaccine constructs.

Round 2

Reviewer 1 Report

The authors responded appropriately to the reviewers' comments.

Reviewer 2 Report

 The manuscript was adequately modified by the authors. I thought that it can be accepted in current form.